# Timeline of changes in spike conformational dynamics in emergent SARS-CoV-2 variants reveal progressive stabilization of trimer stalk with altered NTD dynamics

Sean M Braet[1†], Theresa SC Buckley[1†], Varun Venkatakrishnan[1†], Kim-Marie A Dam[2], Pamela J Bjorkman[2], Ganesh S Anand[1,3,4]*

[1]Department of Chemistry, Pennsylvania State University, University Park, United States; [2]Division of Biology and Biological Engineering, California Institute of Technology, Pasadena, United States; [3]Department of Biochemistry and Molecular Biology, Pennsylvania State University, University Park, United States; [4]The Huck Institutes of the Life Sciences, Pennsylvania State University, University Park, United States

*For correspondence:
gsa5089@psu.edu

†These authors contributed equally to this work

Competing interest: The authors declare that no competing interests exist.

**Abstract** SARS-CoV-2 emergent variants are characterized by increased viral fitness and each shows multiple mutations predominantly localized to the spike (S) protein. Here, amide hydrogen/deuterium exchange mass spectrometry has been applied to track changes in S dynamics from multiple SARS-CoV-2 variants. Our results highlight large differences across variants at two loci with impacts on S dynamics and stability. A significant enhancement in stabilization first occurred with the emergence of D614G S followed by smaller, progressive stabilization in subsequent variants. Stabilization preceded altered dynamics in the N-terminal domain, wherein Omicron BA.1 S showed the largest magnitude increases relative to other preceding variants. Changes in stabilization and dynamics resulting from S mutations detail the evolutionary trajectory of S in emerging variants. These carry major implications for SARS-CoV-2 viral fitness and offer new insights into variant-specific therapeutic development.

## Editor's evaluation

This fundamental and timely study provides insights into the structural dynamics of several relevant mutant forms of SARS-CoV-2 spike protein, including the most recent omicron variant. The hydrogen/deuterium exchange studies provide compelling evidence for the stabilization of the spike stalk in conjunction with increased dynamics of the N-terminal domain, where binding to the ACE2 receptor occurs. These results have profound implications for the development of small molecule inhibitors of the spike protein-ACE2 receptor interaction.

## Introduction

SARS-CoV-2 was first identified in late 2019 and is the causative agent of the ongoing coronavirus pandemic (*Chang et al., 2020*). Extensive efforts have sparked the development of a number of vaccines and therapeutics to mitigate the effects of infection. However, the emergence of numerous variants of concern has created an additional challenge to treatment and prevention efforts. The

**Figure 1.** Strimer modifications and variant mutations. (**A**) The Spike (S) protein trimer (PDB:6VSB) (protomers in dark blue, teal, and green) with receptor-binding domain (RBD), N-terminal domain (NTD), S1/S2 site, fusion peptide, 2P substitutions (985-986), additional 6P substitutions (817, 892, 899, 942), and glycans. (Image created in https://Biorender.com) (**B**) Sequence organization for SARS-CoV-2 S (NTD = N-terminal domain, RBD – receptor binding domain, SD1=subdomain 1, SD2=subdomain 2, FP = fusion peptide, HR1=heptad repeat 1, HR2=heptad repeat 2, T.A.=transmembrane domain, I.T.=intracellular domain). The brackets define the recombinant soluble S used in this study. Furin cleavage site (685) is indicated by arrowhead (**C**) Mutations in Alpha, Delta, and Omicron BA.1 S mapped onto the S structure. Mutations in the NTD, RBD, S1, and S2 domains are represented as blue, green, purple, and yellow dots, respectively, wherever a mutation can be visualized. (**D**) Mutations specific to Alpha, Delta, and Omicron BA.1 S variants.

The online version of this article includes the following source data for figure 1:

**Source data 1.** Mutations in S variants.

periodic emergence of new variants of concern beginning with Alpha, and later including Delta and most recently Omicron BA.1, have contributed to surges in cases worldwide (*Wassenaar et al., 2022*).

SARS-CoV-2 is a member of the family *Coronavirdae* along with other human pathogens including SARS-CoV and MERS (*Corman et al., 2018*). The SARS-CoV-2 virion is enveloped and encapsulates a 30 kb +ssRNA genome that encodes envelope (E) protein, membrane (M) protein, spike (S) protein, as well as 16 non-structural proteins and nine accessory proteins (*Ke et al., 2020*). S, a critical viral protein for SARS-CoV-2 entry that is targeted by neutralizing antibodies, plays a multifunctional role in the infection process and is, therefore, a target for vaccine development (*Martínez-Flores et al., 2021*). S is a glycosylated homotrimer with each monomer consisting of S1 and S2 subunits (*Figure 1A and B*). The S1 domain comprises an N-terminal domain (NTD), a receptor-binding domain (RBD), and two subdomains SD1 and 2, with the RBD mediating the interaction interface with the human ACE2 receptor (*Lan et al., 2020*). The S2 domain includes the S1/S2 and S2 proteolytic cleavage sites as well as a fusion peptide.

S plays three critical roles in facilitating host cell entry: S must bind ACE2, be proteolytically processed, and promote membrane fusion. During the viral entry process, the S is processed by furin protease at the S1/S2 cleavage site either prior to or after S binding to ACE2 receptor. This enables secondary cleavage by a separate protease (commonly transmembrane serine protease 2 (TMPRSS2) or cathepsin) at the S2 site (*Peacock et al., 2021*; *Shang et al., 2020*; *Vankadari, 2020*), which leads to dissociation of S1 and release of the S2 subunit to drive membrane fusion and cellular entry.

Domain-specific investigation of S and its variants have provided insights into the effects of mutations on functionalities in isolation. However, there is a need to address the composite impact of individual variants on S conformation. Altered conformations in mutant S from variants would impact interactions of S with ACE2 (*Raghuvamsi et al., 2021*) and downstream functions.

Due to its key roles in viral host recognition and entry, it is unsurprising that S is a hotspot for mutations in emerging variants (*Tian et al., 2021*). Among the first set of mutations that were detected in S variants during the early phase of the pandemic, D614G emerged as a dominant mutation in 2020 (*Chang et al., 2020*; *Pandey et al., 2021*). One of the striking effects observed in a comparison of wild-type (WT) and D614G S revealed a~50 X enhancement in proteolytic processing by furin (*Gobeil et al., 2021*). The Alpha variant subsequently emerged in September 2020, and in addition to D614G, other mutations localized to NTD (del 69–70, del 144), RBD (N501Y, A570D), the furin-binding site (P681H), and the S2 subunit (T716I, S982A, and D1118H) (*Xia et al., 2021*; *Figure 1C and D*). A more dominant Delta variant, first identified in October 2020 replaced the Alpha variant. The Delta variant S included mutations in the NTD (T19R, G142D, del 156–157, R158G), RBD (L452R, T478K), furin cleavage site (P681R), and the S2 subunit (D950N) (*Tian et al., 2021*). A more recent surge in infection has been due to the dominant Omicron variant (BA.1) first identified in November 2021 and is the variant with the largest number of mutations relative to WT (*Figure 1C and D*; *Liu et al., 2022a*). Notably, the D614G mutation has been conserved across all major variants of concern (*Wassenaar et al., 2022*). Additionally, the P681R mutation found in the Delta S and Omicron BA.1 S has been found to increase pathogenicity and proteolytic processing (*Liu et al., 2022b*; *Saito et al., 2021*), and RBD mutations in variants of concern have been found to increase affinity for the ACE2 receptor (*Han et al., 2022*; *Ozono et al., 2021*). Omicron shows the highest affinity for ACE2 among preceding variants, attributable to several mutations on the RBD (*Kim et al., 2023*).

A defining feature of emerging variants is that each of these became more dominant over prevailing strains, which has been attributed to progressively increased viral fitness (*Liu et al., 2022b*; *Plante et al., 2021*; *Ulrich et al., 2022*). Viral fitness in SARS-CoV-2 has been quantified by competition assays involving the co-infection of two variant strains in airway cell lines and animal models. The variant demonstrating superior viral fitness traits showed higher virus titers post-infection and outcompeted the other variant. From these experiments, it was concluded that D614G S showed greater fitness than WT S (*Plante et al., 2021*), Alpha S showed greater fitness than D614G S (*Ulrich et al., 2022*), and Delta S showed greater fitness than Alpha S (*Liu et al., 2022b*), which likely contributed to new surges in human infections.

Snapshots from single-particle cryo-EM structures of SARS-CoV-2 S trimers have provided structural insights at high resolution (*Cai et al., 2020*; *Duan et al., 2020*; *Walls et al., 2020*; *Zhang et al., 2021*) but do not completely capture all of the interconverting conformations in solution. Multiple conformations showing RBD in 'up' or 'down' orientations have been observed by cryo-EM. In the trimer, these translate into closed- all 'down' or one, two or all 'up' conformations (*Barnes et al., 2020a*). These interchanging conformations in solution highlight the ensemble behavior of S. Dynamics of the S ensemble are fundamental to assessing trimer stability and the role of conformational substates in receptor binding, proteolytic processing, and disease propagation. The ensemble properties have been shown to be critical for ACE2 recognition. The RBDs of S have been reported to bind the ACE2 receptor only in an 'up' conformation (*Barnes et al., 2020a*).

Amide hydrogen deuterium exchange mass spectrometry (HDXMS) is a useful method for probing dynamic breathing motions and conformational ensembles in viral systems (*Costello et al., 2022*; *Lim et al., 2017a*; *Lim et al., 2017b*; *Narang et al., 2021*; *Raghuvamsi et al., 2021*). HDXMS uses $D_2O$ as a probe that labels backbone amides with deuterium dependent upon both solvent accessibility (*Peacock et al., 2018*) and H-bond propensities (*Englander and Kallenbach, 1983*). The labeling reaction can be quenched to probe time scales ranging from second to day with shorter timescales impacted primarily by changes in solvent accessibility and longer timescales assessing changes in H-bonding (*Peacock et al., 2018*). Pepsin proteolysis combined with mass spectrometry provides a readout of deuterium exchange at peptide resolution that can be mapped onto a structure (*Hoofnagle et al., 2003*). This captures the dynamics (>seconds timescale) of the whole ensemble. Furthermore, it offers an ability to resolve more than one slow interchanging conformations (if present) by deconvolution of bimodal distributions of deuterium exchanged mass spectral envelopes (*Hodge et al., 2020*; *Hoofnagle et al., 2003*; *Oganesyan et al., 2018*). Decreased deuterium exchange

reflects protection from solvent and/or enhanced stability and correspondingly; increased exchange reports increased solvent accessibility and/or disorder.

Comparative HDXMS analysis of recombinant WT, D614G, Alpha, Delta, and Omicron BA.1 S variants has allowed us to track changes in intrinsic dynamics across conserved regions of S through the progressively emerging variants of concern. Our results reveal that the timeline of emergence corresponds to an overall stabilizing effect on the S trimer, together with altered dynamics in the NTD and RBD. These loci encompass sites on S showing the greatest differences in deuterium exchange in non-glycosylated peptides common across all variants. Peptides showing differential deuterium exchange identified at the trimeric interface referred to as the stalk region in the rest of the study, report inter-protomer interactions while changes in the NTD and RBD report intra-protomer interactions. Stabilization of the trimer stalk and alteration of NTD dynamics effects are independent, with Delta S achieving near-maximal stabilization as measured by HDXMS in our experimental timescales. Variants showed altered NTD dynamics with Omicron showing the greatest magnitude increases compared to the predecessor variants and WT. These results underscore the importance of stalk stability together with NTD dynamics upon overall S trimer dynamics, with implications for ACE2 recognition, binding, and proteolytic processing.

## Results

### Equilibration at 37°C shifts the S ensemble toward prefusion conformation

Recombinant soluble S constructs have made them more accessible for structural and biophysical research by obviating the need to culture pathogenic SARS-CoV-2 viruses, which require extensive safety procedures and related infrastructure. Engineered S ectodomain constructs show increased expression yields through the ablation of furin protease cleavage site and enhancement of stability through proline substitutions (2P) (*Amanat et al., 2021*) and 6P (hexapro) (*Hsieh et al., 2020*). These engineered constructs also eliminated the need for detergent solubilization by excluding the transmembrane C-terminal segments that are embedded in the lipid bilayer in S assembled on intact SARS-CoV-2 particles (*Barnes et al., 2020b*).

We carried out our HDXMS analysis with a construct containing either 2P or 6P substitutions and with the four amino acid furin cleavage motif (RRAR) substituted with a single alanine (*Amanat et al., 2021*). These were expressed in HEK-293T cells (*Barnes et al., 2020b*). S trimers were purified by nickel nitrilotriacetic acid- agarose (Ni-NTA) and size exclusion chromatography (SEC) as described in the methods. The trimer state was independently verified by cryo-EM analysis (*Barnes et al., 2020a*; *Barnes et al., 2020b*). S is multiply glycosylated with 22 potential N-linked glycosylation sites (*Watanabe et al., 2020*). We confirmed our purified trimeric S was multiply glycosylated and mapped N-linked glycosylation sites by bottom-up proteomics as described in the methods. Of 22 potential N-linked glycosylation sites, we identified 20 sites in WT S, 21 glycosylation sites in D614G and Delta variant S, and 19 glycosylation sites in Omicron BA.1 S (*Supplementary file 1*). WT and variant S in subsequent sections denote either a 2P (*Pallesen et al., 2017*) or 6P (hexapro) (*Hsieh et al., 2020*) as described.

We carried out comparative HDXMS on 2P and 6P-engineered WT S to probe differences in dynamics. 160 non-glycosylated pepsin fragment peptides provided S sequence coverage of 53.1% (*Figure 2—figure supplement 1*). HDXMS was measured at time points ($D_{ex}$ = 1–10 min). A deuterium exchange difference map and hybrid significance testing (*Hageman and Weis, 2019*) (shown in volcano plots *Lau et al., 2021* of the 2P and 6P constructs) revealed no significant differences in deuterium exchange in our experimental timescales ($\Delta$<0.5 Da, p-value = 0.01) (*Figure 2—figure supplement 2*), indicating both 2P and 6P constructs offered a common equivalent baseline for assessing differences across S variants (*Figure 2—figure supplement 1*).

The soluble S trimer constructs have been observed to show sensitivity to cold temperature treatment (*Costello et al., 2022*). Negative stain EM (nsEM) analysis following long-term storage at 4°C revealed heterogeneous S conformations indicative of trimer instability. This heterogeneity was reversed by incubation at 37°C for 3 hr which recovered a well-formed and more homogenous trimeric structure (*Edwards et al., 2021*). It is yet to be determined if this cold sensitivity is an inherent property of the intact S or is relevant only to the S trimer ectodomain constructs.

To test the effects of temperature optimization on the S trimer ectodomain, we compared HDXMS on WT S (2P) treated with and without a 3 hr incubation at 37°C after flash freezing and long-term storage at –80°C. 127 non-glycosylated pepsin fragment peptides provided a primary sequence coverage of 48.4%, and HDXMS was measured at time points ($D_{ex}$ = 1–10 min) (*Figure 2—figure supplement 3*). It should be noted that the pepsin fragmentation of 2P and 6P are not identical due to the differential Pro substitutions (*Figure 2—figure supplement 1*). A deuterium exchange heat map (% RFU) of WT S shown in *Figure 2A* shows higher relative exchange on the outer edges of the trimer compared to the intra-trimer core (*Figure 2A*). Multiple regions of S showed decreased exchange with the greatest magnitude decreases at the trimer interface (*Figure 2B and F*). Decreases were most prominent for peptides in the trimer stalk region of S (peptides 899–913, 988–998, 1013–1021) and other inter-protomer contacts (peptides 553–568 and 32–48) (*Figure 2C–E*, *Figure 2—figure supplement 4*). It should be noted deuterium exchange at certain peptides (for instance, 553–568 (*Figure 2C*) following 37°C incubation (3 hr)) still showed spectral envelopes indicating the presence of more than one conformation.

Resolvable bimodal mass spectral distributions were evident for core helical peptides in loci 899–913 and 988–998 (*Figure 2—figure supplement 5*). Our results are consistent with localized trimer-protomer transitions at the inter-protomer interface shown previously (*Costello et al., 2022*). Furthermore, 37 °C treatment stabilized S trimers, whereas incubation at 4°C prior to deuterium exchange (see methods) resulted in a time-dependent reversal of stabilization ($t_{1/2}$ = 17 hr) as described (*Costello et al., 2022*). We correlate increased stability with decreased deuterium exchange. In the context of peptides at the trimer stalk interface that displays bimodal spectra in our experimental timescales, this decreased exchange can be attributed to ensemble shifts toward a lower exchanging prefusion conformation and stronger inter-protomer contacts. Overall, our results reveal an increase in propensity for the prefusion conformation.

We also observed that replicate measurement of samples maintained close to 0°C for 4 hr between replicates in an automated HDX workflow highlighted expected conformational reversibility. To further examine this effect, we eliminated 0°C incubation of replicate samples by maintaining a constant 20°C incubation of samples throughout the experiment (*Figure 2—figure supplement 6*). These results highlight temperature-sensitive reversible contacts at the edge of a trimer interface core or trimer stalk region. This region encompasses a long central helical segment (987–1031) and a helix flanking the heptad repeats (900-913) (*Walls et al., 2020*). Variants of S were resistant to cold denaturation at 0°C (for 8 hr between first and third technical replicates) (*Figure 2—figure supplement 7*, *Figure 3—figure supplements 4–5*).

## Global conformational changes conferred by the D614G substitution

One of the earliest conserved mutations identified in emergent variants was D614G (*Pandey et al., 2021*) which demonstrated increased viral fitness along with enhanced furin proteolytic cleavage (*Gobeil et al., 2021*). To measure the impact of this mutation upon S dynamics, we compared HDXMS of D614G S with WT S. Comparative HDXMS between D614G and WT S (2P constructs) was carried out using our previously established 37°C temperature incubation (3 hr) treatment to compare equivalent trimer stabilized states. No peptides spanning the D614G mutation site were identified and, therefore, all peptides analyzed were common to both WT and D614G S. 120 non-glycosylated pepsin fragment peptides were identified covering 45.8% of the D614G sequence (*Figure 3—figure supplement 1*).

Relative fractional uptake values for the D614G variant mapped onto an S trimer structure (PDB 6VXX) for $D_{ex}$ = 10 min showed a similar relative deuterium exchange profile to that for WT S. The central stalk region showed lower exchange relative to the peripheral surface accessible regions consistent with it forming the stable core of the trimer (*Figure 3A*). A difference map (D614G minus WT) ($D_{ex}$ = 10 min) (*Figure 3B*) revealed three non-contiguous clusters of peptides distal to the D614G site of mutation, showing the following differences in exchange: (i) Decreased exchange at the trimer interface and (ii) Increased exchange at NTD, and (iii) Increased exchange in heptad repeat segments.

## Decreased exchange at the trimer stalk region in D614G

Deuterium exchange difference plots showed a small subset of contiguous peptides from three regions within the trimer stalk region that showed the largest magnitude protection in deuterium

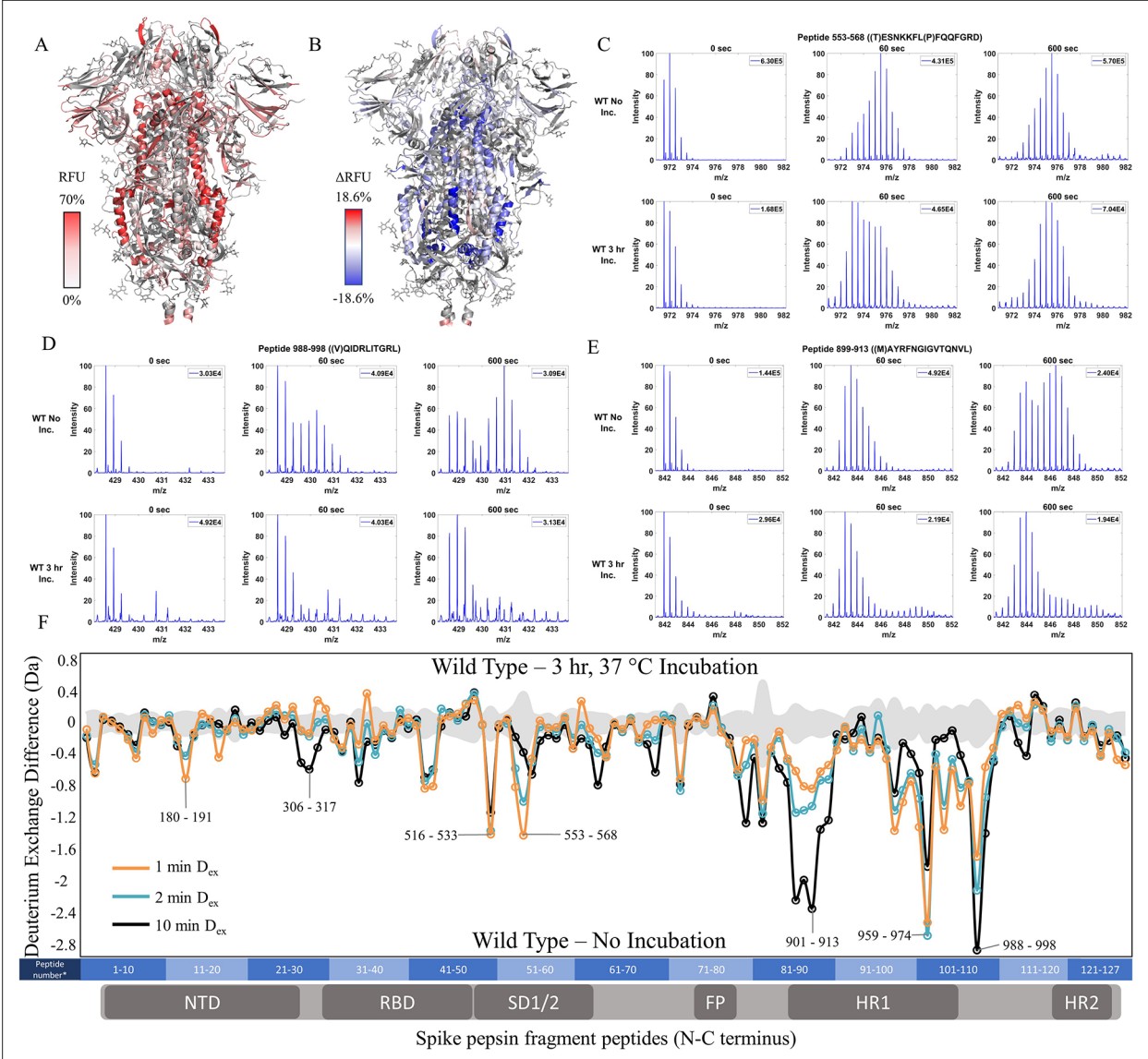

**Figure 2.** Prefusion conformation for wild-type (WT) spike (S) protein favored by 3 hr incubation at 37°C. (**A**) Relative fractional uptake (D$_{ex}$ = 10 min) for unincubated WT S mapped onto an S trimer structure with three 'down' RBDs (PDB ID: 6VXX) (coverage of WT S constructs shown in ***Figure 2—figure supplements 1–2***, differences in deuterium exchange for WT 2P and 6P constructs shown in ***Figure 2—figure supplement 3***). Deuterium exchange heat map gradient of white (0%) -red (70%) as mapped on S structure (PDB ID: 6VXX). (**B**) Differences in deuterium exchange (Δ RFU) (D$_{ex}$ = 10 min) for WT S after a 3 hr incubation at 37°C minus unincubated WT S were mapped onto the S structure (PDB ID: 6VXX). Shades of blue correspond to negative differences in deuterium exchange and shades of red correspond to a positive difference in deuterium exchange. (**C–E**) Stacked mass spectra for WT S peptides 553–568, 899–913, and 988–998 with undeuterated reference spectra, 1 min and 10 min exchange (left to right). For each peptide, the top row shows spectra for unincubated WT S and the bottom row shows spectra for WT S incubated for 3 hr at 37°C. Absolute intensities are indicated at the top right of each spectrum. (**F**) Differences in deuterium exchange (deuterons) mapped at peptide resolution from N to C terminus for WT S incubated for 3 hr at 37°C minus unincubated WT S are shown in difference plots for 1-, 2-, and 10 min exchange. Select peptides showing significant differences in exchange are annotated. Significance was determined by hybrid significance testing (p<0.01, ***Figure 2—figure supplement 4***). Differences are tabulated in ***Figure 2—source data 1*** with corresponding peptide numbers* shown on the x-axis of the difference plot.

The online version of this article includes the following source data and figure supplement(s) for figure 2:

**Source data 1.** Deuterium uptake differences for Incubated WT S minus unincubated WT S.

**Source data 2.** Deuterium uptake differences for WT 6P S minus WT 2P S.

**Source data 3.** Source data for the volcano plot; ***Figure 2***, ***Figure 2—figure supplement 2***.

**Source data 4.** Source data for the volcano plot; ***Figure 2***, ***Figure 2—figure supplement 4***.

**Figure supplement 1.** Comparative HDXMS of wild-type (WT) spike (S) protein 2P and 6P.

*Figure 2 continued on next page*

*Figure 2 continued*

**Figure supplement 2.** Volcano plot analysis of wild-type (WT) 2P versus 6P S.

**Figure supplement 3.** Primary sequence coverage map of pepsin fragment peptides for wild-type (WT) spike (S) protein incubated at 37°C versus unincubated WT S.

**Figure supplement 4.** Volcano plot analysis of wild-type (WT) spike (S) protein 37°C versus WT S unincubated.

**Figure supplement 5.** Deuterium exchanged spectral envelopes for overlapping peptides from two stalk loci in wild-type (WT) spike (S) protein.

**Figure supplement 6.** Effect of ~0°C storage in the course of automated hydrogen deuterium exchange mass spectrometry (HDXMS) on wild-type (WT) trimer peptides.

**Figure supplement 7.** Effect of ~0°C storage in the course of automated hydrogen deuterium exchange mass spectrometry (HDXMS) on deuterium exchange at variant spike (S) protein trimer locus peptides.

exchange ($D_{ex}$ = 1 min) in the D614G variant (*Figure 3F*, *Figure 3—figure supplement 2*). These regions also showed decreased exchange upon a 3 hr incubation at 37°C. Representative peptides 899–913, 988–998 (*Figure 3—figure supplement 3*), and 1013–1021 are reporters for deuterium exchange at the stalk region (*Figure 3—source data 1*). A similar trend with the decreased exchange in this locus with 3 hr incubation at 37°C was seen with D614G S as with WT S (*Figure 3—figure supplements 4–5*). Incubation for 3 hr at 37°C showed a smaller magnitude difference in deuterium exchange overall for D614G compared to WT S (*Figure 3—figure supplements 4–5*). Our results are consistent with the higher stability demonstrated previously for D614G (*Gobeil et al., 2021*; *Yang et al., 2021*) The enhanced stabilization of D614G S is likely a combination of conformational changes and population shifts in the ensemble behavior of S trimers in solution.

## Increased exchange in NTD of D614G S

A minor difference in D614G not found upon 37°C stabilization of WT S was observed in the NTD peptides (peptides spanning regions 243–265 and 306–318) (*Figure 3C*, *Figure 3—source data 1*), each of which showed increased exchange relative to WT S. This revealed that the single point mutation at D614 to glycine, induced long-range allosteric effects that are propagated across the trimer and are associated with both stalk stabilization and increased S1 domain dynamics at the NTD. These were the only loci outside the heptad repeat and across the S1 and S2 domains to show significant differences as shown in a Volcano plot (p<0.01) (*Figure 3—figure supplement 2*) in deuterium exchange between WT and D614G at non-glycosylated and observed peptides. These effects provided a baseline for tracking conformational changes in S in emergent variants with improved fitness. The large conformational changes elicited by the D614G mutation underscore its importance as a highly conserved mutation across emergent variants (*Aleem et al., 2022*).

## Alpha variant S showed increased exchange relative to D614G at both trimer stalk and NTD

We extended our analysis of D614G to emergent variants of concern that each carried this mutation together with multiple other mutations (*Figure 1*). We compared each subsequent variant with its epidemiological predecessor to track changes in deuterium exchange across the timeline of emergence. 45.9% coverage was obtained with 127 non-glycosylated pepsin fragment peptides common to D614G, Alpha, and WT S for a comparative HDXMS ($D_{ex}$ = 1–30 min) analysis of the Alpha S versus D614G S (*Figure 4—figure supplement 1*). Relative fractional uptake for the Alpha S was mapped onto a WT S structure (PDB 6VXX) (*Figure 4A*). Differences in deuterium uptake (Δ RFU) for the Alpha S minus D614G S are mapped onto PDB 6VXX in *Figure 4B*.

The Alpha S compared to D614G S showed lower magnitude differences in deuterium exchange than D614G S compared to WT S showed. In the trimer stalk region for the Alpha S, changes in deuterium exchange in Alpha relative to D614G S were lower in magnitude than in the comparison between D614G and WT S. No perceptible bimodal spectral envelopes indicative of conformational heterogeneity was seen in the trimer stalk region (*Figure 4—figure supplement 3*). This indicated that the bulk of the stabilization of the stalk region in the Alpha S was contributed by the conserved D614G mutation.

Outside the stalk, changes in deuterium exchange were primarily observed at the NTD for common peptides (*Figure 4F*, *Figure 4—figure supplement 2*). Alpha S showed increased exchange at peptides

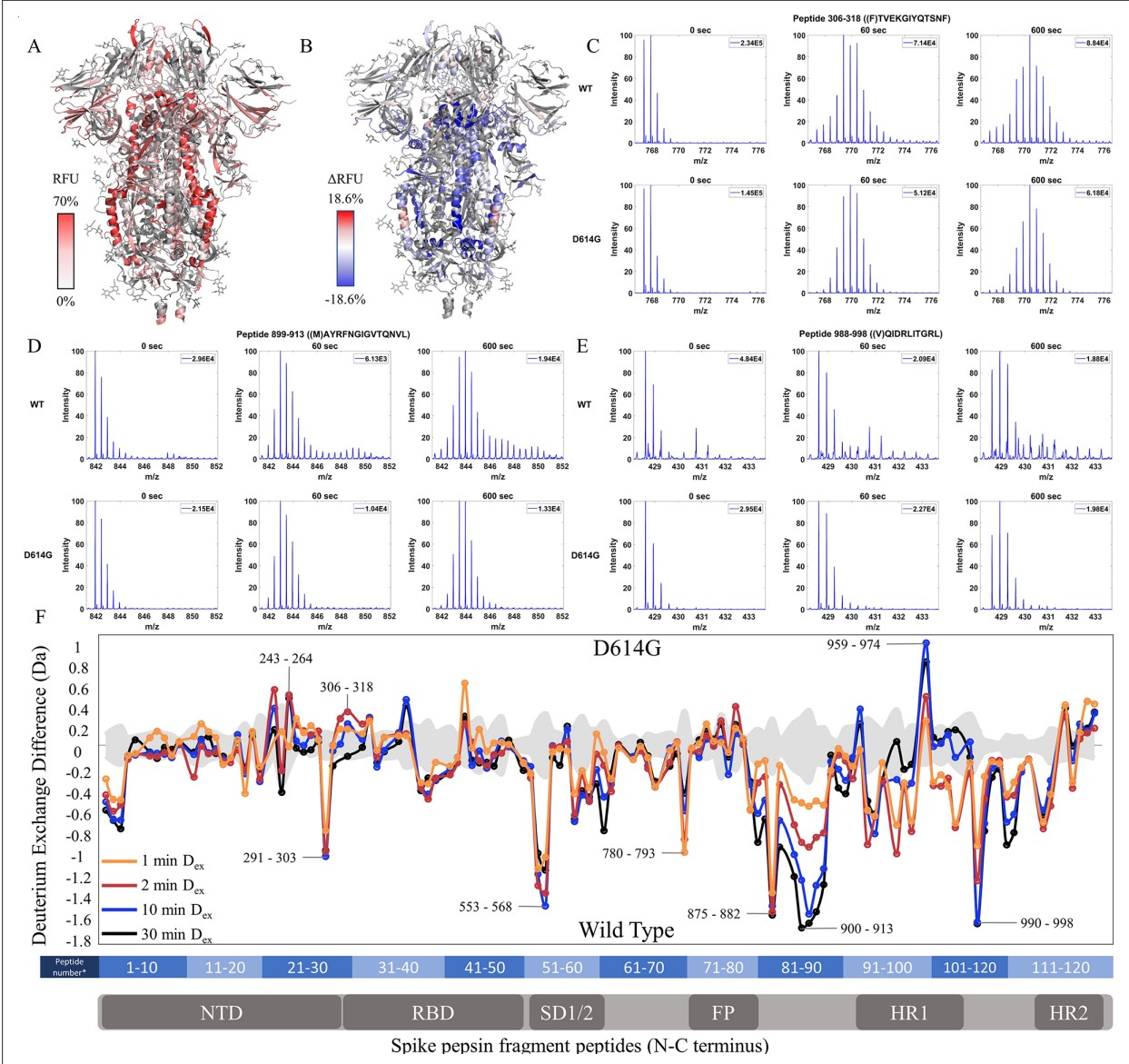

**Figure 3.** The D614G mutation imparts stabilization to spike (S) protein trimer stalk. (**A**) Relative fractional uptake ($D_{ex}$ = 10 min) for D614G S mapped onto an S structure (PDB ID: 6VXX) (coverage maps shown in *Figure 3—figure supplement 1*, differences in deuterium exchange for D614G S incubated at 37°C minus D614G S without incubation is shown in *Figure 3—figure supplement 4*). Deuterium exchange heat map gradient of white (0%) -red (70%) as mapped on S structure (PDB ID: 6VXX). (**B**) Differences in deuterium exchange (Δ RFU) ($D_{ex}$ = 10 min) for D614G S minus wild-type (WT) S were mapped onto an S structure (PDB ID: 6VXX). Shades of blue correspond to negative differences in deuterium exchange and shades of red correspond to a positive difference in deuterium exchange. (**C–E**) Stacked mass spectra for peptides 245–265, 900–913, and 988–998 with undeuterated reference spectra, 1 min and 10 min exchange (left to right). For each peptide, the top row contains spectra for WT S and the bottom row contains spectra for D614G S. Absolute intensities are indicated at the top right of each spectrum. (**F**) Differences in deuterium exchange (deuterons) mapped at peptide resolution from N to C terminus for D614G minus WT S are shown in difference plots for 1-, 2-, 10-, and 30 min exchange. Select peptides showing significant differences in exchange are annotated. Significance was determined by hybrid significance testing (p<0.01, *Figure 3—figure supplement 3*). Back exchange for D614G is estimated in *Figure 3—figure supplement 6*. Differences are tabulated in *Figure 3—source data 1* with corresponding peptide numbers* shown on the x-axis of the difference plot.

The online version of this article includes the following source data and figure supplement(s) for figure 3:

**Source data 1.** Deuterium uptake differences for Incubated D614G S minus WT S.

**Source data 2.** Deuterium uptake differences for Incubated D614G S minus unincubated D614G S.

**Source data 3.** Source data for the volcano plot; *Figure 3*, *Figure 3—figure supplement 2*.

**Figure supplement 1.** Primary sequence coverage map of pepsin fragment peptides for D614G S versus wild-type (WT) spike (S) protein.

*Figure 3 continued on next page*

*Figure 3 continued*

**Figure supplement 2.** Volcano plot analysis of D614G S versus wild-type (WT) spike (S) protein.

**Figure supplement 3.** Mass spectra of overlapping peptides from two stalk loci in D614G S.

**Figure supplement 4.** Hydrogen deuterium exchange mass spectrometry (HDXMS) analysis of incubation effects on D614G S.

**Figure supplement 5.** Primary sequence coverage of pepsin fragment peptides for D614G S incubated at 37°C versus unincubated D614G S.

**Figure supplement 6.** Deuterium back exchange measurements for pepsin fragment peptides from D614G S.

spanning 92–103, 177–191, and 201–264 (*Figure 4F*). Lower magnitude decreases in exchange were observed in the S2 domain, specifically at peptides at the C-terminal end of the heptad repeat one in the 940–975 region (Δ Ex = 0.4–0.8 Da), as well as peptide 553–568 (Δ Ex = 0.6 Da, 30 min), which mediates inter-monomer contacts.

## Delta S shows decreased exchange at both the trimer stalk and NTD

Comparative HDXMS of the Delta S to the Alpha S generated 47.0% coverage was obtained with 123 non-glycosylated pepsin fragmentation peptides common to D614G and the Delta and Alpha S variants (*Figure 5—figure supplement 1*). RFU for $D_{ex}$ = 1 min in the Delta S and differences in exchange for Delta S minus Alpha S (Δ RFU) were mapped onto an S structure (PDB: 6VXX) (*Figure 5A and B*). No perceptible bimodal spectral envelopes indicative of conformational heterogeneity were seen in the trimer stalk region (*Figure 5—figure supplement 3*). Based on volcano plot analyses (p<0.01), insignificant differences in deuterium exchange were observed for other S2 peptides relative to Alpha S (*Figure 5—figure supplement 2*).

Delta S showed decreased exchange relative to the Alpha S for most peptides with decreases primarily in the trimer stalk (*Figure 5F*, *Figure 5—figure supplement 2*), other S2 domain peptides, and the NTD (*Figure 5C–E*). Delta S showed decreased exchange for NTD peptides relative to Alpha S. Decreased exchange was most prominent at peptides spanning 92–103, 177–191, and 200–265. Additional decreases in exchange were observed at 306–317 (Δ Ex = 0.6 Da, 2 min) while the increased exchange was observed at the RBD peptide 407–420 (Δ Ex = 0.4 Da, 2 min). Again, the conserved D614G contributed to the bulk of the stabilization seen in Delta S.

## Omicron BA.1 S retains low exchange at the trimer stalk while showing increased deuterium exchange at NTD peptides

Finally, we compared Omicron BA.1 S to Delta S. 36.4% coverage was achieved with 95 non-glycosylated pepsin fragment peptides common to D614G S (*Figure 6—figure supplement 1*). RFU for $D_{ex}$ = 1 min in Omicron BA.1 S and differences in exchange for Omicron BA.1 S minus Delta S (Δ RFU) were mapped onto a WT S structure (PDB: 6VXX) (*Figure 6A and B*). It should be noted that the Omicron BA.1 S used for analysis was the 6P construct. This was necessitated by poor expression of the Omicron BA.1 S 2P construct. Based on our analysis of WT S, the 2P and 6 constructs showed no differences in HDXMS, without 37°C incubation (*Figure 2—figure supplement 1*).

The Omicron BA.1 S showed slightly decreased exchange at the trimer stalk with the increased exchange at other S2 domain peptides and increased exchange at the NTD (*Figure 6C–F*, *Figure 6—figure supplement 2*). No perceptible bimodal spectral envelopes indicative of conformational heterogeneity were seen in the trimer stalk region (*Figure 6—figure supplement 3*). Notably, the Omicron BA.1 S showed a decrease in the higher exchanging population for stalk peptides suggesting both an impact on inherent trimer stability and ensemble behavior. In S2 domain peptides 968–974 (Δ Ex = 0.9 Da, 2 min), 820–829 (Δ Ex = 0.7 Da, 2 min), and 750–756 (Δ Ex = 0.6 Da, 2 min), increased exchange was observed (*Figure 6F*).

In the NTD, significantly increased exchange was observed in Omicron BA.1 S relative to Delta S. Increases were seen at peptides spanning 177–191, 243–265, and 306–317. Additional increases in exchange were observed for RBD peptide 456–467 (Δ Ex = 0.5 Da, 2 min) and peptides 553–568 (Δ Ex = 0.7 Da, 30 min) and 634–643 (Δ Ex = 0.5 Da, 2 min). Omicron BA.1 S was distinct from the Alpha and Delta S in that it adhered to the continued trend of decreased exchange at the trimer stalk and increased exchange in the NTD that were first observed in D614G S.

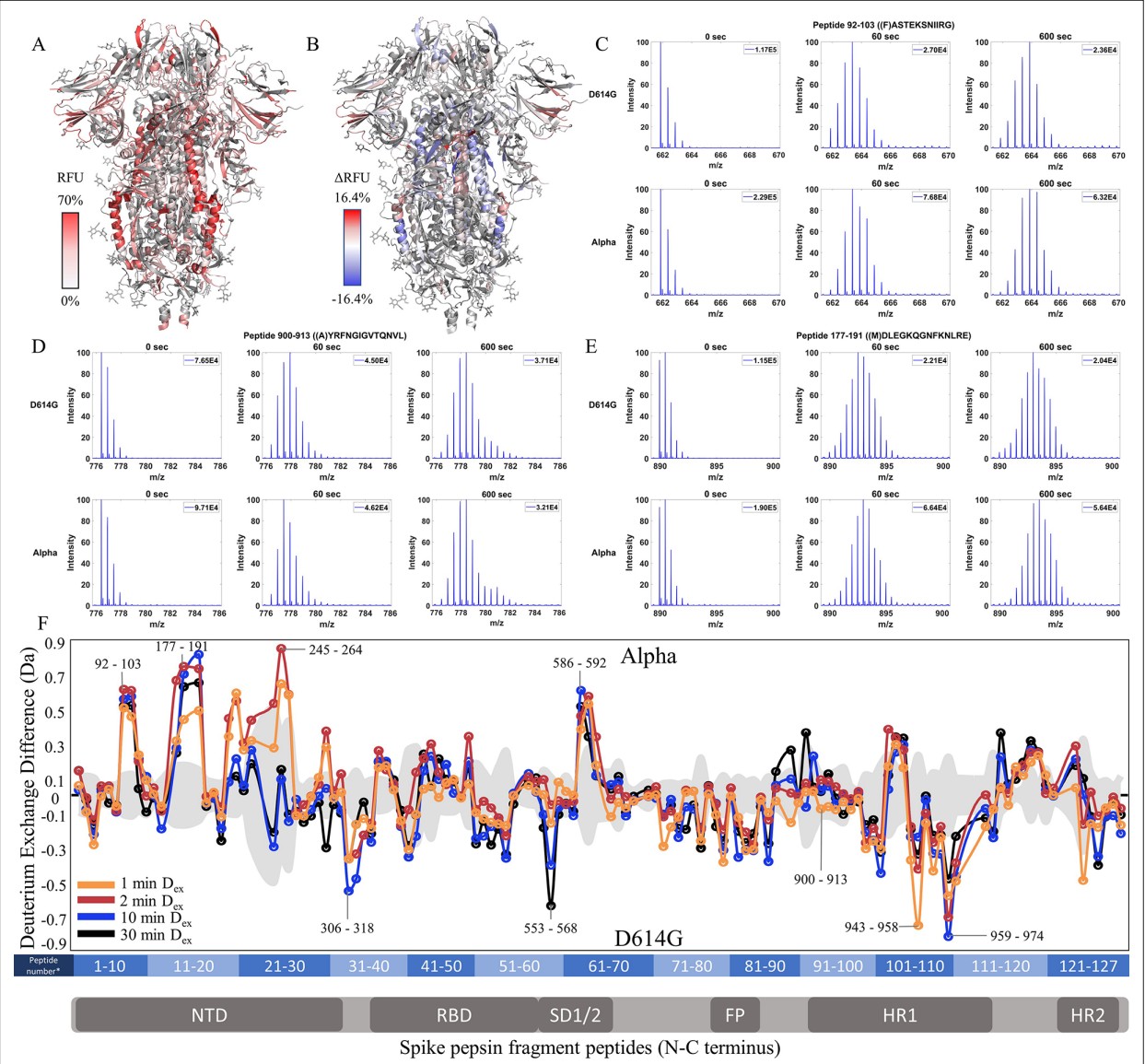

**Figure 4.** Increased N-terminal domain (NTD) dynamics in Alpha S. (**A**) Relative fractional uptake ($D_{ex}$ = 10 min) for Alpha S mapped onto an spike (S) protein structure with three down RBDs (PDB ID: 6VXX) coverage map is shown in *Figure 4—figure supplement 1*). Deuterium exchange heat map (gradient of white (0%) -red (70%) as mapped on S structure (PDB ID: 6VXX). (**B**) Differences in deuterium exchange (Δ RFU) ($D_{ex}$ = 10 min) for Alpha S minus D614G S were mapped onto an S structure (PDB ID: 6VXX). Shades of blue correspond to negative differences in deuterium exchange and shades of red correspond to a positive difference in deuterium exchange. (**C–E**) Stacked mass spectra for peptides 92–103, 177–191, and 900–913 with the undeuterated mass spectral envelope as a reference, 1 min and 10 min exchange (left to right). For each peptide, the top row contains spectra for D614G S and the bottom row contains spectra for Alpha S. Absolute intensities are indicated at the top right of each spectrum. (**F**) Differences in deuterium exchange (deuterons) mapped at peptide resolution from N to C terminus for Alpha S minus D614G S are shown in difference plots for 1-, 2-, 10-, and 30 min exchange. Select peptides showing significant differences in exchange are annotated. Significance was determined by hybrid significance testing (p<0.01, *Figure 4—figure supplement 2*). Differences are tabulated in *Figure 4—source data 1* with corresponding peptide numbers* shown on the x-axis of the difference plot.

The online version of this article includes the following source data and figure supplement(s) for figure 4:

**Source data 1.** Deuterium uptake differences for Alpha S minus D614G S.

**Source data 2.** Source data for the volcano plot; *Figure 4*, *Figure 4—figure supplement 2*.

**Figure supplement 1.** Primary sequence coverage map of pepsin fragment peptides for Alpha S versus D614G S.

**Figure supplement 2.** Volcano plot analysis of Alpha S versus D614G S.

**Figure supplement 3.** Mass spectral envelopes for overlapping peptides from two stalk loci in Alpha S.

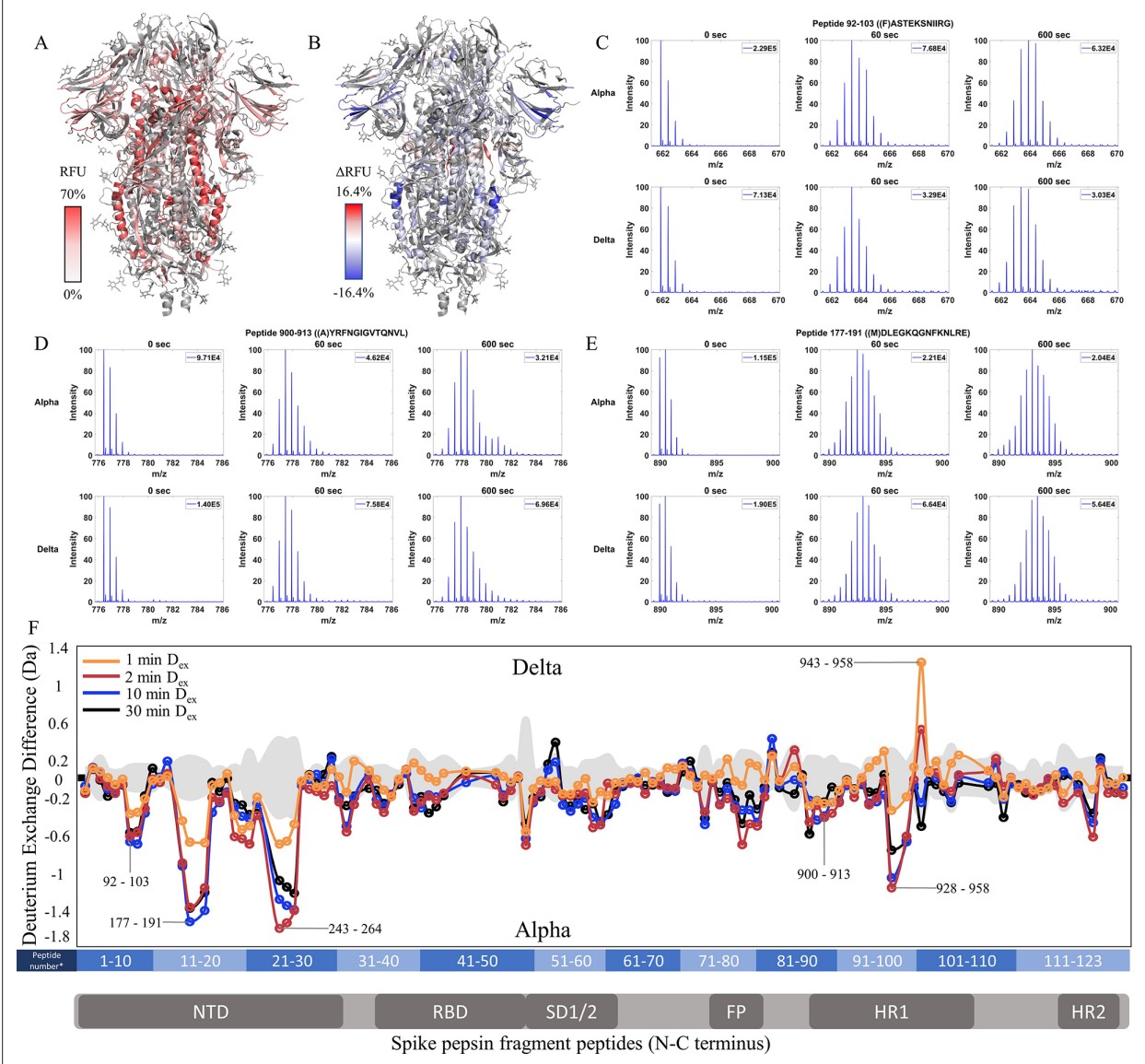

**Figure 5.** Increased stability at trimer stalk and decreased N-terminal domain (NTD) dynamics in Delta S. (**A**) Relative fractional uptake ($D_{ex}$ = 10 min) for Delta S mapped onto a wild-type (WT) spike (S) protein structure with three down receptor-binding domain s (RBDs) (PDB ID: 6VXX) coverage maps shown in *Figure 5—figure supplement 1*). Deuterium exchange heat map (gradient of white (0%) -red (70%) as mapped on S structure (PDB ID: 6VXX). (**B**) Differences in deuterium exchange (ΔRFU) ($D_{ex}$ = 10 min) for Delta S minus Alpha S were mapped onto a WT S structure (PDB ID: 6VXX). Shades of blue correspond to negative differences in deuterium exchange and shades of red correspond to positive differences in deuterium exchange. (**C–E**) Stacked mass spectra for peptides 92–103, 177–191, and 900–913 with undeuterated reference spectra, 1 min and 10 min exchange (left to right). For each peptide, the top row contains spectra for Alpha S and the bottom row contains spectra for Delta S. Absolute intensities are indicated at the top right of each spectrum. (**F**) Differences in deuterium exchange (deuterons) mapped at peptide resolution from N to C terminus for Delta S minus Alpha S are shown in difference plots for 1-, 2-, 10-, and 30 min exchange. Select peptides showing significant differences in exchange are annotated. Significance was determined by hybrid significance testing (p<0.01, *Figure 5—figure supplement 2*). Differences are tabulated in *Figure 5—source data 1* with corresponding peptide numbers* shown on the x-axis of the difference plot.

The online version of this article includes the following source data and figure supplement(s) for figure 5:

**Source data 1.** Deuterium uptake differences for Delta S minus Alpha S.

**Source data 2.** Source data for the volcano plot; *Figure 5*, *Figure 5—figure supplement 2*.

**Figure supplement 1.** Primary sequence coverage map of pepsin fragment peptides for Delta S versus Alpha S.

**Figure supplement 2.** Volcano plot analysis of Delta variant S versus Alpha variant S.

**Figure supplement 3.** Mass spectra from overlapping peptides from two stalk loci in Delta S.

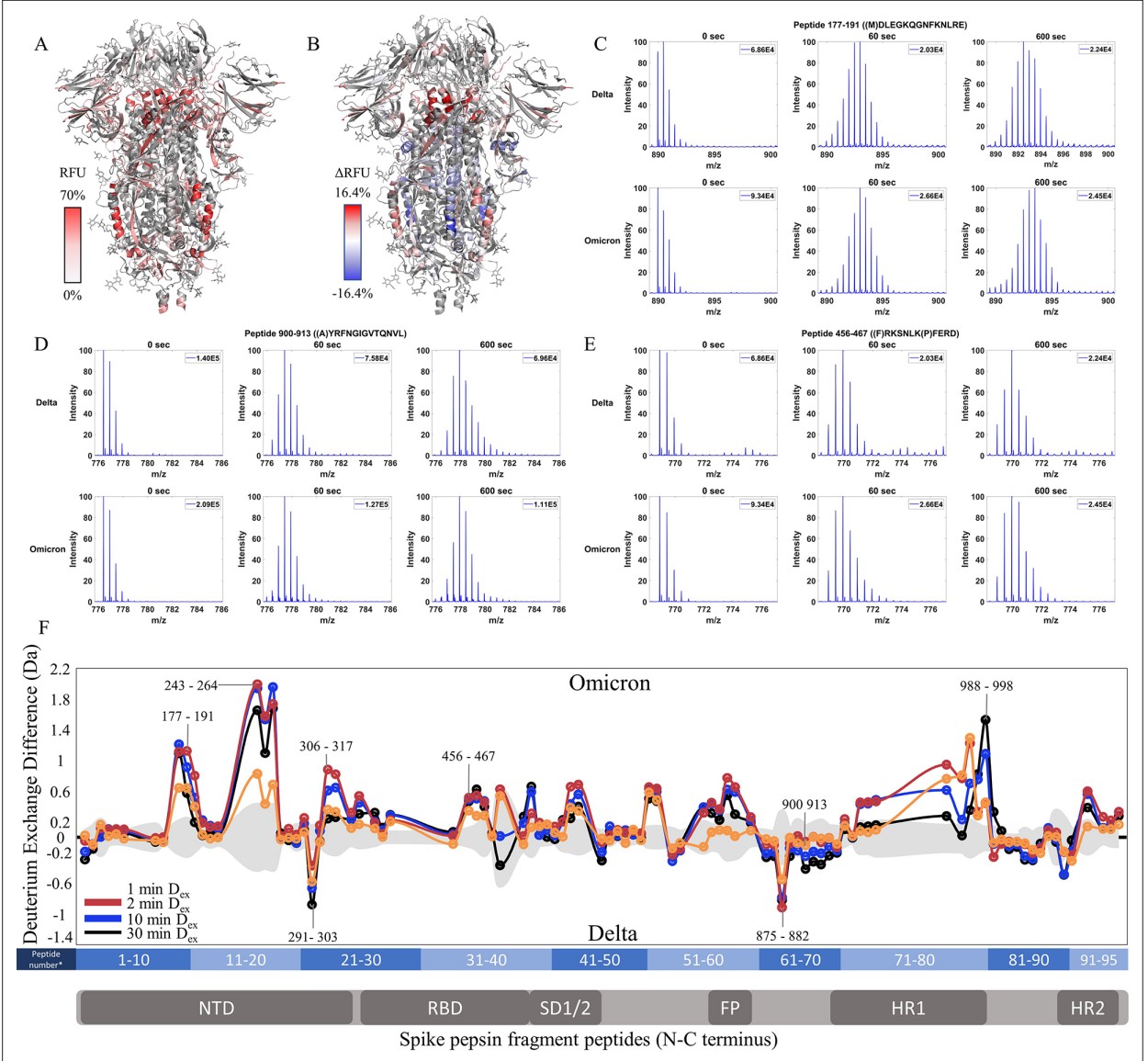

**Figure 6.** Enhanced trimer stability and N-terminal domain (NTD) dynamics in Omicron BA.1 S. (**A**) Relative fractional uptake ($D_{ex}$ = 10 min) for Omicron BA.1 S mapped onto a wild-type (WT) spike (S) structure with three down receptor-binding domains (RBDs) (PDB ID: 6VXX) coverage maps shown in *Figure 6—figure supplement 1*). Deuterium exchange heat map (gradient of white (0%) -red (70%) as mapped on WT S structure (PDB ID: 6VXX). (**B**) Differences in deuterium exchange ($\Delta$RFU) ($D_{ex}$ = 10 min) for Omicron BA.1 S minus Delta S were mapped onto a WT S structure (PDB ID: 6VXX). Shades of blue correspond to negative differences in deuterium exchange and shades of red correspond to a positive difference in deuterium exchange. (**C–E**) Stacked mass spectra for peptides 92–103, 177–191, and 900–913 with undeuterated reference spectra, 1 min, and 10 min exchange (left to right). For each peptide, the top row contains spectra for Delta S and the bottom row contains spectra for Omicron BA.1 S. Absolute intensities are indicated at the top right of each spectrum. (**F**) Differences in deuterium exchange (deuterons) mapped at peptide resolution from N to C terminus for Omicron BA.1 S minus Delta S are shown in difference plots for 1-, 2-, 10-, and 30 min exchange. Select peptides showing significant differences in exchange are annotated. Significance was determined by hybrid significance testing (p<0.01, *Figure 6—figure supplement 2*). Differences are tabulated in *Figure 6—source data 1* with corresponding peptide numbers* shown on the x-axis of the difference plot.

The online version of this article includes the following source data and figure supplement(s) for figure 6:

**Source data 1.** Deuterium uptake differences for Omicron BA.1 S minus Delta S.

**Source data 2.** Source data for the volcano plot; *Figure 6*, *Figure 6—figure supplement 2*.

**Figure supplement 1.** Primary sequence coverage map of pepsin fragment peptides for Omicron S versus Delta S.

**Figure supplement 2.** Volcano plot analysis of Omicron variant S versus Delta variant S.

**Figure supplement 3.** Mass spectral envelopes from overlapping peptides from two stalk loci in Omicron S.

## Discussion

We report two uncorrelated effects of mutations upon the conformational dynamics of variant S. The first major trend we observed was a stabilization of the trimer stalk region. This was followed by altered NTD and RBD dynamics that emerged in later variants. We noted a stabilizing effect of 37°C temperature treatment on the stalk region of S, however, this treatment had no effect on NTD/RBD dynamics, indicating that these trends are not allosterically coupled.

### Progressive stabilization of trimer stalk

The first prominent trend observed in our analysis of S variants was the progressive stabilization of the S trimer. Using HDXMS, we were able to observe the ensemble behavior of the S trimer. WT S is sensitive to exposure to ~0–4°C (*Costello et al., 2022*), and the ensemble behavior was a heterogeneous mixture of different prefusion conformations. This is explained by the observation that the sequences of three representative peptides examined in the stalk region adopt an amphipathic helical fold (*Figure 7—figure supplement 1*). The propensity to undergo cold denaturation can be attributed to the hydrophobic interactions at the stalk region that contribute to stability at the trimer interface (*Costello et al., 2022*; *Edwards et al., 2021*; *Privalov, 1990*). Incubation at 37°C shifted the ensemble to favor the prefusion conformation.

Conformational heterogeneity has been observed in other S trimers from HIV-1, MERS, and SARS-CoV (*Derking and Sanders, 2021*; *Kirchdoerfer et al., 2018*; *Pallesen et al., 2017*) highlighted analogous stem loci to comprise a conformational dynamic switch between distinct prefusion conformations. To overcome the conformational heterogeneity in SARS-CoV and MERS S trimers, two consecutive proline substitutions were shown to confer an improved expression of prefusion trimers (*Kirchdoerfer et al., 2018*; *Pallesen et al., 2017*). These mutations were extended to SARS-CoV-2 S. The SARS-CoV-2 2P constructs were also found to be more immunogenic, making it a preferred construct for vaccine development (*Hsieh et al., 2021*; *Lien et al., 2021a*; *Lien et al., 2021b*). Mutations introduced on a 2P background were screened for improved expression and four additional proline substitutions were identified. These formed the basis for the hexapro (6P) (*Hsieh et al., 2020*) substitution construct also widely used for structural and biophysical characterization of S. These independent effects of structure-guided mutations suggest an evolutionary advantage of S trimer stabilization in SARS-CoV-2.

The first mutation to confer a large stabilization on the S trimer stalk was D614G. Comparing WT S to D614G S we observed substantial stabilization (up to 1.7 Da decreases in exchange). The decreased exchange is a consequence of a shift in the ensemble toward a higher proportion of prefusion conformation and, to a lesser degree, intrinsic stabilization of the prefusion conformation. Progressive stabilization was observed in subsequent variants (three representative peptides in *Figure 7A and B*) with the highest stabilization occurring in Delta S and Omicron S (*Figure 7—figure supplements 2 and 3*). After the initial increase in stabilization from D614G, increases in subsequent variants were lower in magnitude (*Figure 7B*). It is notable that a single point mutation (D614G) which is conserved across nearly all recent emergent SARS-CoV-2 variants of concern (*Pandey et al., 2021*), had a significantly larger impact on stability than other any other mutations studied. Comparative analysis of peptides encompassing mutations in variant S with D614G S further validated D614G as the most prominent contributor to enhanced trimer stabilization (*Supplementary file 2*). D614G is a lynchpin S mutation and HDXMS of S demonstrates some mutations have a disproportionate impact on protein conformation and stability.

The D614G mutation breaks a salt bridge between D614 of one protomer and either T859 or K854 on a neighboring protomer. Since this salt bridge is proximal to the trimer stalk, this provides a possible explanation for the large increase in stalk stability. These observations are also consistent with increased S packing density and decreased premature shedding on viruses containing D614G S (*Zhang et al., 2020*). Increased stability would improve downstream function and viral fitness which is observed in competition assays (*Plante et al., 2021*). Since the subsequent variants displayed further increases in stability, it is likely that this effect is also associated with increased viral fitness (*Liu et al., 2022b*; *Ulrich et al., 2022*).

### Altered NTD dynamics conferred by variant mutations

In addition to increased stability, we observed variable dynamics at the NTD and RBD. While D614G resulted in the most prominent increase in stability, only limited effects were observed at the NTD.

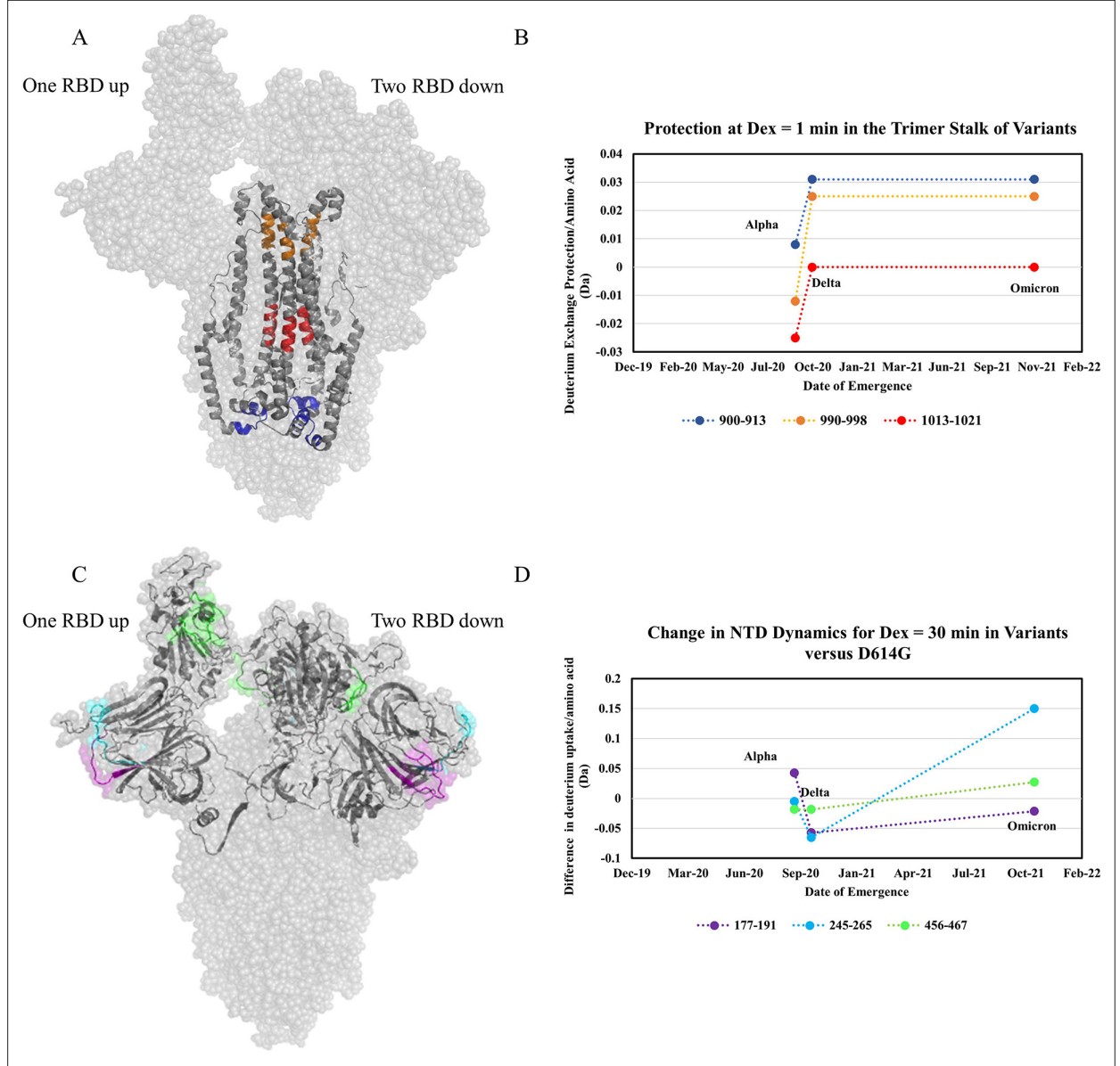

**Figure 7.** Increased trimer stability and altered N-terminal domain (NTD) dynamics correlate with the timeline of emergence. (**A**) Trimer stalk peptides 900–913 (blue), 990–998 (orange), and 1013–1021 (red) mapped onto a wild-type (WT) spike (S) protein structure (PDB ID: 7TGX) (helical wheel analysis of stalk peptides shown in **Figure 7—figure supplement 1**). (**B**) Protection in trimer stalk peptides in S variants compared to D614G S plotted as protection per amino acid versus date of emergence at $D_{ex}$ = 1 min. (**C**) NTD and RBD peptides showing increased dynamics in the timeline of variant emergence mapped onto a 1 RBD 'up' WT S structure (PDB ID: 7TGX). Peptides 177–191, 245–265, and 456–467 are shown in purple, cyan, and green, respectively. (**D**) Changes in deuterium uptake for NTD and receptor-binding domain (RBD) peptides in variant S compared to D614G S at $D_{ex}$ = 30 min are plotted as a change in deuterium uptake versus date of emergence (additional plots in **Figure 7—figure supplement 2**). Uptake plots for representative peptides in all variants are shown in **Figure 7—figure supplement 3**.

The online version of this article includes the following figure supplement(s) for figure 7:

**Figure supplement 1.** Hydrophobic interactions maintain trimer core at stalk region.

**Figure supplement 2.** Plot of protection in spike (S) protein trimer stalk peptides as function of timeline of emergence.

**Figure supplement 3.** Deuterium uptake plots comparing all spike (S) protein variants.

Comparing D614G S to Alpha S, however, revealed a notable increase in exchange at NTD peptides (*Figure 7C*). Delta S showed a decrease in NTD dynamics relative to Alpha S, but Omicron BA.1 S showed the highest overall NTD and RBD dynamics (*Figure 7C and D*).

These changes are likely associated with the role of NTD and RBD dynamics in promoting ACE2 recognition and correspondingly increased viral entry across successive variants (*Qing et al., 2021*). N501Y (*Figure 1C*, in green), found in Alpha and Omicron BA.1 S, has been found to enhance ACE2:S interactions 16-fold (*Tian et al., 2021*). Increased NTD dynamics might facilitate RBD 'up' transitions. A570D (*Figure 1C*, in purple), found in Alpha S, has been implicated in modulating RBD dynamics to enhance ACE2 interactions (*Yang et al., 2021*). This, coupled with mutations at the S-ACE2 interface would then facilitate increased efficacy of SARS-CoV-2 host entry. It should be noted that HDXMS reports an average deuterium exchange for an ensemble of conformations when the rates of conformational interconversion are faster relative to rates of exchange (EX2 kinetics) at relevant experimental conditions (*Hoofnagle et al., 2003*). HDXMS results mapped onto a single end-state cryo-EM structure are useful in identifying dynamic loci on S. However, this reports an average conformational change across protomers rather than capturing all conformations of protomers within a trimer and, therefore, precludes descriptions of RBD up and down transitions that cannot be resolved.

The S variants have been shown to have an increased propensity for the RBD up conformation which is required for ACE2 recognition (*Yang et al., 2022*; *Yurkovetskiy et al., 2020*). An allosteric network between the NTD and RBD has also been established (*Ray et al., 2021*). Our work indicates that this increased RBD-up propensity may be facilitated by increased dynamics at the NTD since we also observed this trend in emerging variants. This provides an additional conformational basis for the observed increase in variant fitness *McCallum et al., 2020* by priming the S for ACE2 interaction and consequent cellular entry.

In summary, our results localize the impacts of mutations on conformational dynamics to the trimer stalk region and NTD. This provides an analysis of S trimer conformational dynamics through the timeline of variant emergence. Every successive variant S displays a progressively increased stability at the central stalk region in the S2 subunit. D614G S showed a large magnitude stabilization at the stalk region with minor increases in exchange at the NTD. Alpha S showed no differences at the stalk region but showed increases in NTD and RBD dynamics. Delta S showed more stabilization at the stalk region and decreased NTD dynamics. Omicron BA.1 S showed even greater stabilization of the stalk region together with increased NTD and RBD dynamics. Overall, these results suggest that while near-maximal trimer stalk stability has been achieved, emerging variants continue to show progressive increases in NTD and RBD dynamics. Changes in stabilization and dynamics resulting from specific S mutations detail the evolutionary trajectory of S in emerging variants. This provides a basis for progressively enhanced viral fitness and carries major implications for S evolution and therapeutic development.

## Materials and methods
### Expression and purification of SARS-CoV-2 S

Expression and purification of SARS-CoV-2 S ectodomains were performed as previously described (*Barnes et al., 2020a*; *Barnes et al., 2020b*). SARS-CoV-2 S constructs were composed of residues 16–1206 of the early SARS-CoV-2 isolate (GenBank MN985325.1), Alpha variant (GISAID EPI_ISL_601443), Delta variant (GenBank QWK65230.1), or Omicron variant (BA.1) (GISAID EPI_ISL_9845731) with the following stabilizing mutations: 2P (*Pallesen et al., 2017*) or 6P (*Hsieh et al., 2020*), the furin cleavage site mutated to Ala, a C-terminal TEV protease site (GSG-RENLYFQG), foldon trimerization motif (GGGSG-YIPEAPRDGQAYVRKDGEWVLLSTFL), 8x-His tag (G-HHHHHHHH), and AviTag (GLNDIFEAQKIEWHE). All S constructs were expressed using the Expi293T transient transfections system (GIBCO). S trimers from clarified transfected cell supernatants were purified over HisTrap High Performance columns (Cytiva), followed by size-exclusion chromatography (SEC) using a Superose 6 increase 10 300 columns (Cytiva). Fractions corresponding to S trimers were collected and concentrated in 10% glycerol TBS (20 mM Tris, 150 mM NaCl, pH 8.0) then flash-frozen in liquid nitrogen and stored at –80°C. Downstream purification was completed within 12 hr of the transfected cell harvest to maximize the quality of trimeric, well-folded S trimers.

## Bottom-up proteomics and glycan profiling

Recombinant S variants were digested with trypsin overnight. Samples were separated by RP-HPLC using a Thermo Scientific EASY-nLC 1200 UPLC system connected to a Thermo Scientific PepMap C18 column, 15 cm × 75 µm over a 90 min 5–25%, 15 min from 40–95% gradient (A: water, 0.1% formic acid; B: 80% acetonitrile, 0.1% formic acid) at 300 nL/min flow rate. The samples were analyzed on the Thermo Scientific Orbitrap Eclipse Tribrid mass spectrometer using the DDA FT HCD MS2 method. FT MS1 was acquired at resolution settings of 120 K at m/z 200 and FTMS2 at the resolution of 30 K at m/z 200.

The Thermo Scientific Proteome Discoverer 2.5 software with the Byonic search node (Protein Metrics) was used for glycopeptide data analysis and glycoform quantification. Data were searched against a database containing the Uniprot/SwissProt entries of the model proteins with/out common contaminants and 57 human plasma glycans with a 1% FDR criteria for protein spectral matches. The peptide spectra were also manually validated to confirm the identification of glycosylation sites.

## Deuterium exchange

Labeling buffer was prepared by diluting 20 X PBS in $H_2O$ in $D_2O$ (99.9%). 3 µL of the sample were added to 57 µL of labeling buffer for a final labeling concentration of 90.16%. Deuterium labeling was carried out for 1, 2, 10, 30, and 100 min at 20°C using a PAL-RTC (Leap) autosampler. During automated HDXMS experiments protein samples were stored at 0°C and stability was assessed by staggering technical replicates. This allowed us to evaluate the suitability of automated HDXMS workflows for evaluating S. It should be noted that WT S experiments were conducted without 0°C storage due to observed variability during automated runs and Delta S was collected on two separate days eliminating 0°C storage. After labeling, equivalent volumes of labeling reaction and prechilled quench solution (1.5 M GdnHCl, 0.25 M TCEP) was added to bring the reaction to pH 2.5. Reaction conditions are summarized in *Supplementary file 3*.

## Mass spectrometry and peptide identification

Approximately 8–10 pmol of the sample were loaded onto a BEH pepsin column (2.1 × 30 mm) (Waters, Milford, MA) in 0.1% formic acid at 75 µL/min. Proteolyzed peptides were trapped in a C18 trap column (ACQUITY BEH C18 VanGuard Pre-column, 1.7 µM, Waters, Milford, MA). Peptides were eluted in an acetonitrile gradient (8–40%) in 0.1% formic acid on a reverse phase C18 column (AQUITY UPLC BEH C18 Column, Waters, Milford, MA) at 40 µL/min. All fluidics were controlled by nano-ACQUITY Binary Solvent Manager (Waters, Milford, MA). Electrospray ionization mode was utilized, and ionized peptides were sprayed onto an SYNAPT XS mass spectrometer (Waters, Milford, MA) acquired in HDMS[E] Mode. Ion mobility settings of 600 m/s wave velocity and 197 m/s, transfer wave velocity were used with collision energies of 4 and 2 V were used for trap and transfer, respectively. High collision energy was ramped from 20 to 45 V while a 25 V cone voltage was used to obtain mass spectra ranging from 50 to 2000 Da (10 min) in positive ion mode. A flow rate of 5 µL/min was used to inject 100 fmol/µL of [Glu[1]]-fibrinopeptide B ([Glu[1]]-Fib) as lockspray reference mass.

Peptides of WT and SARS CoV-2 variant S were identified through independent searches of mass spectra from the undeuterated samples in two steps. First, peptides common to WT and variant S were identified from a database containing the amino acid sequence of WT and D614G S using PROTEIN LYNX GLOBAL SERVER version 3.0 (Waters, Milford, MA) software in HDMS[E] mode for non-specific protease cleavage. Search parameters in PLGS were set to 'no fixed or variable modifier reagents' and variable N-linked glycosylation.

Deuterium exchange was quantitated using DynamX v3.0 (Waters, Milford, MA) with cutoff filters of: minimum intensity = 2000, minimum peptide length = 4, maximum peptide length = 25, minimum products per amino acid = 0.2, and precursor ion error tolerance <10 ppm. Three undeuterated replicates were collected for WT and variant S, and the final peptide list includes only peptides that fulfilled the above-described criteria and were identified independently in at least 2 of the 3 undeuterated samples.

In the second step, the workflow was repeated to identify peptides unique to S variants. Pepsin fragment peptides from each variant S were identified from a database containing the amino acid sequence of the corresponding variant S. Deuterium exchange in these peptides was analyzed using DynamX 3.0 with identical parameters described above.

## Hydrogen deuterium exchange analysis

The average number of deuterons exchanged in each peptide was calculated by subtracting the centroid mass of the undeuterated reference spectra from each deuterated spectra. Peptides were independently analyzed for quality across technical replicates. Relative deuterium exchange and difference plots were generated by DynamX v3.0. The data for the mass spectra were acquired from DynamX v3.0 and plotted in MATLAB 2022a, The MathWorks Inc, Natick, MA, USA. Relative deuterium exchange plots are reported as RFU which is the ratio of exchanged deuterons to possible exchange deuterons. Back exchange estimates were determined by RFU values from a 24 hr labeling experiment and are shown in *Figure 3—figure supplement 6*. Deuterons (*Lau et al., 2021*) was used to generate the hybrid volcano plots using hybrid significance testing (p<0.01) (*Hageman and Weis, 2019*). The mass spectrometry proteomics data will be deposited to the ProteomeXchange Consortium via the PRIDE partner repository.

## Acknowledgements

Startup funding from the Pennsylvania State University (PSU) to GSA. We thank Rosa Viner (Thermo Scientific, San Jose, CA) for S glycan analysis. We thank Susan Marqusee (University of California Berkley, CA) for discussions and help with revision.

## Additional information

### Funding
No external funding was received for this work.

### Author contributions
Sean M Braet, Data curation, Software, Formal analysis, Investigation, Visualization, Methodology, Writing – original draft, Writing – review and editing; Theresa SC Buckley, Data curation, Formal analysis, Investigation, Visualization, Methodology, Writing – original draft, Writing – review and editing; Varun Venkatakrishnan, Data curation, Formal analysis, Investigation, Methodology, Writing – original draft, Writing – review and editing; Kim-Marie A Dam, Resources, Investigation, Methodology, Writing – review and editing; Pamela J Bjorkman, Conceptualization, Resources, Supervision, Investigation, Writing – review and editing; Ganesh S Anand, Conceptualization, Resources, Formal analysis, Supervision, Funding acquisition, Investigation, Visualization, Methodology, Writing – original draft, Project administration, Writing – review and editing

### Author ORCIDs
Sean M Braet http://orcid.org/0000-0003-3782-2262
Theresa SC Buckley http://orcid.org/0009-0005-2897-9630
Pamela J Bjorkman http://orcid.org/0000-0002-2277-3990
Ganesh S Anand http://orcid.org/0000-0001-8995-3067

### Decision letter and Author response
Decision letter https://doi.org/10.7554/eLife.82584.sa1
Author response https://doi.org/10.7554/eLife.82584.sa2

## Additional files

### Supplementary files
• Supplementary file 1. Glycosylation profile of SARS-CoV-2 S variants. N-linked glycans were identified by mass spectrometry. The number of glycans identified at each site and an example glycan are reported.

• Supplementary file 2. HDXMS analysis of mutated peptides. Differences between variants and D614G S for mutated peptides reported in Da for $D_{ex}$ = 1, 2, 10, and 30 min exchange. Mutations sites are bolded and underlined.

• Supplementary file 3. Summary of HDXMS conditions. Experimental conditions and coverage for HDXMS analysis of WT and variant S.

• MDAR checklist

## Data availability

We have made raw files, ProteinLynx Global Server 3.0 search outputs, and Dynamx files used for HDXMS analysis available through ProteomeXchange. The dataset can be found in the PRIDE repository with identifier PXD040717.

The following dataset was generated:

| Author(s) | Year | Dataset title | Dataset URL | Database and Identifier |
|---|---|---|---|---|
| Anand GS, Braet S | 2023 | Timeline of changes in spike conformational dynamics in emergent SARS-CoV-2 variants reveal progressive stabilization of trimer stalk with altered NTD dynamics | http://proteomecentral.proteomexchange.org/cgi/GetDataset?ID=PXD040717 | ProteomeXchange, PXD040717 |

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
