## [Editor Report]

This fundamental and timely study provides insights into the structural dynamics of several relevant mutant forms of SARS-CoV-2 spike protein, including the most recent omicron variant. The hydrogen/deuterium exchange studies provide compelling evidence for the stabilization of the spike stalk in conjunction with increased dynamics of the N-terminal domain, where binding to the ACE2 receptor occurs. These results have profound implications for the development of small molecule inhibitors of the spike protein-ACE2 receptor interaction.

---

## [Decision Letter]

**Decision letter after peer review:**

Thank you for submitting your article "Timeline of changes in spike conformational dynamics in emergent SARS-CoV-2 variants reveal progressive stabilization of trimer stalk and enhanced NTD dynamics" for consideration by *eLife*. Your article has been reviewed by 3 peer reviewers, one of whom is a member of our Board of Reviewing Editors, and the evaluation has been overseen by a Reviewing Editor and Amy Andreotti as the Senior Editor. The reviewers have opted to remain anonymous.

While all the reviewers found this manuscript to be timely and of great interest, there are several comments, which are compiled below.

One of the reviewers raised several technical concerns, particularly related to the heterogeneity of the ensemble, that would need to be addressed in the revisions. Please refer to the specific comments from reviewer #2.

In addition, the reviewers found some of the figures hard to read and suggested the following improvements

1) Please put the residue numbers on the X-axes in all the figures, as without this it is very hard to relate the peptide sequences to the plots. This is particularly problematic in figures 2-S3, 3-S3, 3-S5 which also do not have the protein motifs indicated on the X-axis.

2) Several of the supplemental figures, for example, Figure 2—figure supplement 2 and Figure 7—figure supplement 1, should be prepared with higher resolution. In some of them, it is very difficult to discern the text, due to the poor image resolution and small fonts.

3) The protein numbering appears to be inconsistent between the figures. For example, the main figure 2 presents data on peptides 899-913 and 988-998, but in figure 2-supplement 1 that numbering is different, corresponding to peptides ending at positions 916 and 1001. If there is an established numbering system for the Spike protein then all the presented data should be numbered accordingly.

4) The authors should consider including a summary table for the experimental conditions.

*Reviewer #2 (Recommendations for the authors):*

1) The study makes several comparisons across different variants. The main figures are only useful for assessing differences between 2 variants at a time. I think the paper would benefit from showing a few individual peptide uptake plots for key regions of interest. That way it is easy to readily compare the magnitude of the changes across all of the variants within a single plot.

2) Figures 2-S3, 3-S3, 3-S5 are presented but have no x-axis labels or any other way of indicating which regions are different. These need to be revised to include amino acid sequence information or at the very least highlight and label the regions that are of interest. In their current form, they provide no useful information.

3) The authors should include a summary table of the conditions and experiments as outlined in the community recommendations article (PMID: 31249422). This will help alleviate other uncertainties I mentioned including the method for establishing significance and defining the number of replicates.

4) The protein numbering appears to be inconsistent. For example, the main figure 2 presents data on peptides 899-913 and 988-998 but in figure 2-supplement 1 that numbering is different, corresponding to peptides ending at positions 916 and 1001. If there is an established numbering system for the Spike protein then all the presented data should be numbered accordingly.

5) Several of the supplemental figures, for example, Figure 2—figure supplement 2, should be prepared with higher resolution. For some of them, it is very difficult to discern the text because of the poor image resolution.

*Reviewer #3 (Recommendations for the authors):*

The only suggestion I have is that it is impossible to relate the peptide sequences to the plots in the figures. Please put the residue numbers on the X-axes. It's great that the structural motifs are also included, but without any residue numbers, the data are difficult to interpret. Other than that, this is a beautiful paper!

---

## [Author Response]

While all the reviewers found this manuscript to be timely and of great interest, there are several comments, which are compiled below.

We thank the reviewers for their acknowledgement of the significance of our work.

One of the reviewers raised several technical concerns, particularly related to the heterogeneity of the ensemble, that would need to be addressed in the revisions. Please refer to the specific comments from reviewer #2.In addition, the reviewers found some of the figures hard to read and suggested the following improvements1) Please put the residue numbers on the X-axes in all the figures, as without this it is very hard to relate the peptide sequences to the plots. This is particularly problematic in figures 2-S3, 3-S3, 3-S5 which also do not have the protein motifs indicated on the X-axis.

We acknowledge the need to be able to access the peptide motifs and sequence in all figures. In the interest of space on the figure, we have indexed all peptides in clusters of ten below the plot. An accompanying table of peptides will enable the reader to correlate the peptide with the plots and access the deuterium exchange difference for each individual peptide in the plot. We have contrasted the weight of the plot curve to make it easier for a reader to access individual peptides. This has been standardized for all figures and supplementary figures and should make it a lot easier for the reader to connect the plot to the peptide sequences.

2) Several of the supplemental figures, for example, Figure 2—figure supplement 2 and Figure 7—figure supplement 1, should be prepared with higher resolution. In some of them, it is very difficult to discern the text, due to the poor image resolution and small fonts.

We have increased the font size of the text and enhanced resolution of all figures.

3) The protein numbering appears to be inconsistent between the figures. For example, the main figure 2 presents data on peptides 899-913 and 988-998, but in figure 2-supplement 1 that numbering is different, corresponding to peptides ending at positions 916 and 1001. If there is an established numbering system for the Spike protein then all the presented data should be numbered accordingly.

We understand the inconsistency observed by the reviewer which is due to Figure 2-Supplement 1 showing a comparison deuterium exchange in the 2P and 6P constructs of WT S. The numbering for these 2 constructs are different due to the absence of certain peptides in the 6P construct due to Pro substitutions (817, 892, 899 and 942). Figure 2- S1 describes the difference in model S protein constructs. To improve consistency, we have integrated Figure 2—figure supplements 1 and 3 into a composite Figure 2—figure supplement 1.

We have included a sentence guiding readers to connect Figure 2 with the relevant Supplemental figure 1 for comparing the deuterium exchange in 2P and 6P constructs, and the coverage map (Figure 2—figure supplement 3) for mapping differences upon temperature treatment.

4) The authors should consider including a summary table for the experimental conditions.

We have included Table S3 that summarizes the experimental conditions.

Reviewer #2 (Recommendations for the authors):1) The study makes several comparisons across different variants. The main figures are only useful for assessing differences between 2 variants at a time. I think the paper would benefit from showing a few individual peptide uptake plots for key regions of interest. That way it is easy to readily compare the magnitude of the changes across all of the variants within a single plot.

In the revised manuscript, Figure 7—figure supplement 3 compares all variants and WT for specific peptides of interest.

2) Figures 2-S3, 3-S3, 3-S5 are presented but have no x-axis labels or any other way of indicating which regions are different. These need to be revised to include amino acid sequence information or at the very least highlight and label the regions that are of interest. In their current form, they provide no useful information.

We have included key peptide labels and a table corresponding to the peptide numbers in the revised manuscript.

3) The authors should include a summary table of the conditions and experiments as outlined in the community recommendations article (PMID: 31249422). This will help alleviate other uncertainties I mentioned including the method for establishing significance and defining the number of replicates.

We have included a summary table (Table S3).

4) The protein numbering appears to be inconsistent. For example, the main figure 2 presents data on peptides 899-913 and 988-998 but in figure 2-supplement 1 that numbering is different, corresponding to peptides ending at positions 916 and 1001. If there is an established numbering system for the Spike protein then all the presented data should be numbered accordingly.

We have addressed this (please see response to point 1 of reviewer 2).

5) Several of the supplemental figures, for example, Figure 2—figure supplement 2, should be prepared with higher resolution. For some of them, it is very difficult to discern the text because of the poor image resolution.

We have generated high resolution figures and these are included with the revised manuscript.

Reviewer #3 (Recommendations for the authors):The only suggestion I have is that it is impossible to relate the peptide sequences to the plots in the figures. Please put the residue numbers on the X-axes. It's great that the structural motifs are also included, but without any residue numbers, the data are difficult to interpret. Other than that, this is a beautiful paper!

We thank the reviewer. We have included peptide numbers that correspond to source data.